# Motivations and Attitudes Toward Further Education: A Cross-Sectional, Descriptive Predictive Study

**DOI:** 10.3390/nursrep15060190

**Published:** 2025-05-29

**Authors:** Ivana Sušilović, Marija Ljubičić, Tatjana Matijaš, Ivana Bokan, Mario Marendić

**Affiliations:** 1Department of Surgery, Division of Thoracic and Vascular Surgery, University Hospital of Split, 21000 Split, Croatia; ivana.susilovic6@gmail.com; 2University Department of Health Studies, University of Split, 21000 Split, Croatia; tmatijas@ozs.unist.hr (T.M.); ivanabokan11@gmail.com (I.B.); 3Department of Health Studies, University of Zadar, 23000 Zadar, Croatia; mljubicic@unizd.hr; 4Health School, 21000 Split, Croatia

**Keywords:** nursing, nursing education, motivation, nursing student, professional development, academic success, predictors

## Abstract

**Background:** Educational requirements in healthcare are constantly evolving, and understanding nurses’ motivations toward continuing education is critical to designing nursing programs, developing workforce strategies, and ensuring better healthcare outcomes. **Objectives**: This study aimed to examine the relationships among nurses’ personal factors, motivations, and attitudes toward further education. **Methods:** We conducted a cross-sectional study involving 526 employed nurses. Based on their intention to enroll in studies, the nurses were divided into two groups: those who intended to enroll (n = 276) and those who did not intend to enroll (n = 250). We used the Work Preferences Inventory and the Attitudes and Educational Intentions Scale to assess motivations and attitudes toward further education. The multivariate analysis included linear and logistic regression to assess associations between variables. **Results:** Nurses who intended to enroll in nursing studies had higher intrinsic motivation than nurses who did not. Strong negative associations were found between job challenges and older age (β = −0.68), while length of service was positively correlated (β = 0.46). A lower level of education had a negative effect on overall work motivation (β = −0.15) and attitudes toward future education (β = −0.09). Nurses with higher intrinsic challenge motivation (OR = 1.07) and a positive attitude toward further education (OR = 1.17) were more likely to study nursing. **Conclusions:** Intrinsic motivation, experience, and a positive attitude toward career advancement influence nurses’ intentions to continue their education. To further motivate nurses, it is crucial to improve working conditions, offer advancement opportunities, and foster a culture that values their contributions and growth.

## 1. Introduction

Healthcare professionals are essential for carrying out or mediating many of the healthcare system’s intensive work activities [1,2]. The prominent role of healthcare workers is to promote health and effective healthcare, and it depends on the number and suitable composition of healthcare workers as well as the necessary resources and motivations to perform the assigned tasks [2].

Nurses have the largest role in the healthcare system [3]. According to data from 2020, there are 41,276 registered nurses in Croatia, of which around 32,440 are employed in the healthcare system [4,5]. Only 26% of these nurses have a bachelor’s degree, and around 12% have higher education (master’s or PhD) [4,5]. So, the highest number of nurses in Croatia work in the healthcare system as secondary nursing school graduates.

The healthcare system is constantly changing due to medical science and technological development. Furthermore, there is an increasing need for high-quality care for an aging population; more chronic and malignant diseases, disabilities, and palliative care [6,7] require educated staff. Nurses must be educated to care for an aging population with declining mental and physical health who are becoming increasingly diverse, engage in new professional roles, adapt to new environments and technologies, function in a changing policy environment, and cooperate with other professionals. A skilled workforce can raise the standard of care and improve the healthcare system as a whole [3,8]. Studying nurses already working at a higher level of education than the current one enables them to acquire the latest knowledge and skills they need to meet the challenges of modern practice [9]. Further training opens up opportunities for advancement, such as specialization or taking on management positions, and contributes to self-confidence and motivation [10]. Although educational requirements in healthcare are constantly changing, understanding the motivation of nurses to pursue further education remains important for planning nursing program design, developing workforce development strategies, and ensuring better healthcare outcomes [9].

The most common reasons for advancing the academic education of nurses are the desire for more knowledge, motivation from close people, the desire for a better salary, and advancement at the workplace [11,12]. Shahhosseini and Hamzehgardeshi found that professional skills and updating information were the most common motives, while lack of support was the main obstacle to continuing the education of nurses [13]. Parolisi found that job requirements, advancement in the workplace, and personal and professional growth are the reasons for continuing education from non-BSN RN (registered nurse) to BSN (Bachelor of Science in Nursing), and the problems that prevented students from advancing were family obligations, time and energy to return to school, and lack of support from colleagues [14,15].

There are obstacles for working nurses to earn a higher level of education, such as difficult work–life balance, insufficient time, high financial cost, and the belief that more post-licensure education is not valuable [6]. Regardless, further efforts are needed to increase the number of nurses with higher levels of education.

Motivation is the driving force behind engagement in nursing education and practice [16]. It can be intrinsic motivation, based on greater interest and satisfaction (e.g., personal satisfaction), or extrinsic motivation, based on the expected outcome (e.g., awards and recognition). Considering all of the above, we formulated the following four research questions: (i) Are there any significance differences in intrinsic, extrinsic, and overall motivations between nurses who intend and who do not intend to enroll in further education? (ii) Will nurses’ personal factors (including age, gender, family environment, children, years after finishing high school, workplace, and years of work) significantly predict intrinsic, extrinsic, and overall motivations among nurses? (iii) Will nurses’ personal factors (including age, gender, family environment, children, years after finishing high school, workplace, and years of work) significantly predict attitudes toward further education among nurses? and (iv) Are there any associations between nursing study enrollment and nurses’ personal factors (including age, gender, family environment, children, years after finishing high school, workplace, and years of work)?

## 2. Materials and Methods

### 2.1. Study Design

This study used a cross-sectional, descriptive predictive research design. The study was conducted between April and June 2021 in Croatia. The data were collected through online surveys using the snowball method. Google Forms platform was used to make the online survey. The survey was conducted via social networks (Facebook, WhatsApp) in groups whose users are nurses, with the option of sending a link to contacts and nurses’ colleagues. Participants were reminded four times every two weeks by sharing a link via the Facebook social network, while the link was only sent once via WhatsApp contacts to nurses’ colleagues. The aim of the repeated reminders by sharing the questionnaire link was to collect as many responses as possible from potential participants. To prevent duplicate responses, only one submission per IP address was allowed.

### 2.2. Participants

Convenience sampling was used. The inclusion criteria were completed nursing training and working in the nursing profession. We excluded retired nurses and nurses not employed in the nursing profession. Based on this criterion, 46 participants were excluded. The final sample comprised 526 nurses. To answer the survey question “Do you intend to enroll in further nursing education?” The participants had two response options: Yes/No. According to their answer option, the nurses were divided into two study groups: those who intended to enroll in studies (n = 276) and those who did not (n = 250).

### 2.3. Data Collection Tool

The measuring instrument used in this study was a structured questionnaire consisting of three parts designed to assess nurses’ motivations and attitudes toward continuing education.

The first part included 12 questions related to the general characteristics of the participants, such as gender, age, marital status, level of education, place of residence, as well as inquiries about their place of employment and work status.

The second part of the questionnaire consisted of 30 statements that evaluated motivation based on work preference [17]. The Work Preference Inventory (WIPI) consists of 30 items, with 15 items measuring intrinsic motivation (enjoyment scale, 10 items and enjoyment scale, 5 items) and 15 items measuring extrinsic motivation (outward scale, 7 items and compensation scale, 8 items) [17]. Quantitative measurements of the WIPI scale are obtained using a 5-point Likert scale, where participants circle one digit to express the degree of their personal attitudes toward each statement. The possible answers are: 1—I completely disagree, 2—I partially disagree, 3—I cannot decide, 4—I partially agree, and 5—I completely agree. The maximum score for this section is 150, with a higher score indicating greater motivation. According to the original research, the internal consistency of the questionnaire is satisfactory, with a Cronbach’s alpha of 0.61 for extrinsic motivation and 0.74 for intrinsic motivation [17]. In this study, the internal consistency between all 30 statements was α = 0.80, indicating high reliability [18].

The third part of the questionnaire was developed by the first and last authors based on a literature review [15,19,20,21,22,23,24,25,26,27,28,29,30], as well as the clinical experience of the last author. To assess content relevance, three independent experts in nursing with extensive professional experience reviewed the items. Each expert evaluated the clarity and relevance of the items, and their feedback was used to refine the questionnaire. Given the small number of reviewers, this process should be interpreted as a preliminary assessment of content validity [31]. This Attitudes and Educational Intentions Scale contained 12 questions regarding the participants’ perceptions and attitudes toward potential obstacles to continuing education. The questions in this part of the questionnaire individually examined the attitudes of the participants: 1—I do not have enough time to continue my education, 2—I want to continue my education because of a better salary, 3—I gave up studying a long time ago, 4—Secondary medical school is not enough to meet the challenges of the job, 5—It is not financially worthwhile for me to continue my education, 6—I have other priorities (e.g., family), besides education, 7—I am motivated to continue my education, 8—When I finish my education, my education degree will not be recognized, 9—I want to continue my education, to keep my job, 10—When I finish my education, my education degree will be recognized, 11—I can learn something new if I continue my education, 12—I do not want to do difficult jobs such as bathing patients. This section also used a 5-point Likert scale, where participants circled one digit to indicate the extent to which each statement applied to them. The possible responses were: 1—does not apply to me at all, 2—does not apply to me in part, 3—neither applies to me nor does it not apply, 4—applies to me in part, and 5—applies to me completely. The minimum score was 12, while the highest score was 60. A higher score indicated a positive attitude toward continuing education. In this study, the internal consistency between all 12 statements was α = 0.64, indicating satisfactory reliability [18].

### 2.4. Ethical Considerations

The study was conducted in accordance with the ethical standards of the Declaration of Helsinki. This study was approved by the Ethics Committee of the University Department of Health Studies, University of Split (2181-228-07-21-0010). At the beginning of the questionnaire, the participants gave their consent. All participants were familiar with the purpose of the survey. The survey was explained to them in detail in the instructions for participation at the beginning of the questionnaire. Voluntary and anonymous participation was emphasized. By completing the online questionnaire, participants gave their consent to the study. The expected time to complete the questionnaire was approximately 10 min, and participants were able to complete the online survey at their convenience.

### 2.5. Statistical Analysis

The normality of the distribution was tested using the Kolmogorov–Smirnov test. Cronbach’s alpha was used to check the internal consistency of the questionnaires. We conducted a small pilot study (with 10 participants) to check the comprehensibility of the questionnaire.

Absolute numbers and percentages were used to describe categorical variables. The median and interquartile range were used for numerical variables. The chi-square test and Mann–Whitney test were used to analyze the differences between groups of nurses.

Linear regression models were used to analyze the relationship between sociodemographic characteristics, motivational work orientations, and attitudes toward continuing education. The outcome variables were motivations and attitudes toward continuing education. The predictors were sociodemographic variables, attitudes toward doctoral studies in nursing, and a positive attitude toward continuing education.

Logistic regression was used to predict enrolment in a nursing degree program. The outcome variables were enrolment in a nursing degree program. The predictors were sociodemographic variables, attitudes toward doctoral studies in nursing, and intrinsic and extrinsic motivations.

Statistical analysis was performed using SPSS 25.0 (IBM, Armonk, NY, USA). Statistically significant values were those with *p* < 0.05.

## 3. Results

### 3.1. Sociodemographic Data of Nurses

A total of 572 Croatian participants took part in the study. The sociodemographic characteristics of nurses are shown in Table 1. There were no differences between groups in gender distribution (χ^2^ = 2.00; *p* = 0.157), living environment (χ^2^ = 0.05; *p* = 0.818), and family environment (χ^2^ = 2.85; *p* = 0.092), while nurses who did Not Intend to Enroll in Nursing Studies (NIENS) had a higher percentage of having children (74.0%; χ^2^ = 21.91; *p* < 0.001). Nurses who did Intend to Enroll In Nursing Studies (IENS) were somewhat younger (Mdn = 35.0; IQR = 14.0; Z= −7.77; *p* < 0.001) and had a lower level of education (χ^2^ = 9.75; *p* = 0.021). The IENS group had shorter lengths of service (Mdn = 16.0, IQR = 16.0; Z = −7,33; *p* < 0.001) as well time since finishing high school (Mdn = 14.0, IQR = 17.0; Z= −7.42; *p* < 0.001). In both groups, a dominant number of nurses worked at hospitals (χ^2^ = 11.18; *p* = 0.004). A negative attitude toward doctoral nursing studies was recorded in both groups, but it was more pronounced in the NIENS group (70.0%) than in the IENS group (52.2%; χ^2^ = 17.47; *p* < 0.001). A higher percentage tried to enroll in studies before, noticed in the IENS group (88.8%; χ^2^ = 33.98; *p* < 0.001) (Table 1).

### 3.2. Differences Between Study Groups in Work Motivations and Attitudes Toward Future Education

Nurses who intended to enroll in future studies had significantly higher intrinsic motivation (Z = −3.57; *p* < 0.001) and overall motivation (Z = −2.70; *p* = 0.007). They were more likely to perceive nursing education as challenging (Z = −3.87; *p* < 0.001) and enjoyable (Z = −2.24; *p* = 0.025), in comparison to those who did not intend to advance their studies. There was no statistically significant difference in extrinsic motivation between the two groups. However, there was a significant difference in compensation (Z = −2.41; *p* = 0.016), one dimension of extrinsic motivation (Table 2).

### 3.3. Association Between Work Motivations and Intent to Enroll in Nursing Studies

Regression analysis showed that older age decreased work challenges (β = −0.68; *p* = 0.004), while longer length of service increased it (β = 0.46; *p* = 0.007). Male nurses reported lower enjoyment (β = −0.14; *p* = 0.002) and motivation (β = −0.12; *p* = 0.005). Education was a significant predictor of the outward domain (β= −0.12; *p* = 0.006), challenge (β= −0.11; *p* = 0.017), and overall work motivation (β = −0.15; *p* < 0.001). Working at a hospital negatively contributed to intrinsic motivation (β= −0.17; *p* = 0.002) and the perception of work as challenging (β = −0.15; *p* = 0.005) and enjoyable (β= −0.15; *p* = 0.006). A positive attitude toward further education was a significant predictor for all areas and overall motivation. All independent variables explained 25–39% of the variance (adjusted R^2^ = 0.25–0.39) (Table 3).

Logistic regression showed several characteristics to be associated with nursing study enrollment. Nurses with a high school education (OR = 2.30; *p* = 0.005) and those who had previously attempted to enroll (OR = 3.89; *p* < 0.001) had higher odds of study enrollment compared to university-educated nurses. Additionally, nurses with a positive attitude toward doctoral studies (OR = 1.78; *p* < 0.001), greater intrinsic challenge motivation (OR = 1.07; *p* = 0.036), and a positive attitude toward continuing education (OR = 1.17; *p* < 0.001) were more likely to enroll in nursing studies (Table 4).

## 4. Discussion

This study aimed to assess nurses’ motivations and attitudes toward continuing education and their intentions to pursue further studies in nursing. Nurses with lower levels of education tended to be less motivated, found their work less challenging, and enjoyed it less, while those with higher education and more experience found their work to be more rewarding and challenging, leading to greater motivation and a positive attitude toward further education and career advancement. Older age was strongly negatively associated with work challenges, while length of service was positively associated. This suggests that the perception of challenge decreases with age, as confidence in the ability to perform tasks increases with experience [32]. On the other hand, a longer period of employment often entails more responsibility and more complex tasks, which increases challenges in the workplace [33]. Our research revealed that half of the employed nurses wanted to continue their education. Nurses who had already enrolled in a study program and those with a positive attitude toward doctoral studies were more likely to pursue further education in the future. This suggests that nurses show an interest in further education and professional development, which may indicate a desire for career advancement or the acquisition of new skills and knowledge in the nursing field [34]. However, most had completed secondary education or an undergraduate degree in nursing. Recognizing the need for continuing education to meet all of the challenges of the profession and to maintain professional and personal growth and development is important to develop and support employed nurses. In addition, continuing education is the path to competent and professional nursing.

Logistic regression analysis showed that a positive attitude toward education, a positive attitude toward doctoral studies, a higher level of intrinsic motivation, and a lower level of education predicted enrollment in nursing studies. Our findings are significant because they demonstrate that nurses who aspire to advance in their nursing careers are intrinsically motivated. Intrinsic motivation is a key characteristic of successful professionals, as it enables them to persistently achieve their goals and remain resilient when faced with obstacles [35]. Our findings also confirm that a positive attitude toward education and higher education is important for academic success [36]. It is essential for policymakers to promote a positive attitude toward education to enhance the educated workforce needed in nursing. A recent study, unlike ours, which focused on employed nurses, did not identify a direct impact of intrinsic motivation on career development [37].

It is essential to address various obstacles, such as recognition of completed education and the need for higher salaries. We believe that nurses have a desire to acquire new knowledge and pursue education despite the challenges present in their work environment [38]. Nurses planning to participate in this research were generally younger, had lower education levels, and had less work experience and time since graduating from high school. This finding indicates that employed nurses often negatively oppose pursuing higher education. Many struggle to commit to further studies or obtain a doctorate; some even consider leaving the profession altogether [11,39,40,41].

Research shows that nurses who pursue higher levels of education, such as the Master of Nursing Science (MNSc) program, engage in a reflective process to assess their current career status and future goals. These nurses’ main motivations are to make a difference and contribute positively to their communities and society as a whole [9].

Our research has shown that, in Croatia, employed nurses have a negative attitude toward a doctorate in nursing. Although employed nurses who intended to enroll in nursing studies had a somewhat more positive attitude toward doctoral studies, employed nurses had a negative attitude toward doctoral studies. Bokan et al. found a more positive attitude toward PhDs among undergraduate and graduate nursing students [11]. More than 65% of nursing students believed that PhDs were necessary for nurses. This difference in attitudes toward PhDs could be explained by employment, as most undergraduate students were unemployed [11]. A possible reason for this is the systematization of jobs in Croatia, which does not follow the educational system, so nurses still do not perceive higher education as important. The existence of a doctorate is very important for the nursing profession. Nurses with a doctorate are qualified to occupy leadership positions in the healthcare system, improve evidence-based nursing care, teach at nursing universities, and conduct scientific research [6].

The slow increase in the number of nurses with doctorates is very important for the nursing profession [6]. These nurses play a key role as educators and contribute to the education of future nurses. They are instrumental in researching health-related issues, such as the impact on health outcomes and the pursuit of health equity [6]. This can have a negative impact on society in general.

The slow growth of PhD-prepared nurses is a significant concern for the profession [6]. Many nurses still overlook its importance, which requires further examination in future research. The employed nurses who intended to enroll in nursing studies in our study had a more positive attitude toward future education. The increasingly complex requirements of an aging population, as well as the challenges associated with the rapid progress of science and technology, demand more advanced competencies from nurses that they acquire through continuous education and higher education. Policymakers should consider this finding and create policies that contribute to a positive attitude toward higher education, systematize jobs, recognize educational qualifications, and involve nurses in the development of health policies. This finding also needs to be considered due to our finding that a positive attitude toward future education and doctoral studies in nursing are positively associated with all domains and overall work motivation. The results concerning male participants should be carefully viewed because of their small representation in the sample. Research has shown that the belief that additional post-licensure education is not valuable and a lack of peer support are issues that have prevented practicing nurses from obtaining a higher level of education [6,15].

In addition, nurses in the IENS group had a higher level of intrinsic motivation, a higher level on the challenge and enjoyment subscale, and compensation as a subscale of extrinsic motivation. Motivation greatly influences students’ success and the goals they set for themselves. Intrinsically motivated students cope with more challenges and are more persistent in their desire to achieve their goals [42]. Our findings are similar to previous studies that found that the desire for more knowledge, which can be considered intrinsic motivation, a desire for better pay, and advancement in the workplace, as external motivations, were the most common reasons for later education among nurses [11,12,13,15].

Working in a hospital negatively contributed to intrinsic motivation and the perception of work as challenging and enjoyable. Here, one could ask the question of nurses’ satisfaction with hospital employees; however, our research was not designed to answer this question. However, research has shown that motivation can decline due to the loss of the original purpose of nursing, and loss of motivation leads to nurse burnout and higher turnover rates [37,43]. Job satisfaction for nurses could be increased by improving their career development. Career growth also positively affects the intention to stay in the current organization. In addition, support from the organization and management is of utmost importance for continuing education in the form of time-sharing, recognition, and validation of competencies acquired through education, and support and advice from colleagues [19].

There has been limited research on how nurses’ attitudes toward continuing education affect their motivation for further education. Future studies should include additional factors, such as salary, job position, and other relevant social and economic indicators.

### Limitations

Our study has limitations that should be acknowledged. Firstly, as a cross-sectional design, it does not establish a cause-and-effect relationship. Additionally, the low response rate among employed nurses may have impacted the significance of the findings. Another limitation is the small number of male participants, which limits the generalizability of gender-related findings. Some studies have indicated that male individuals may be less inclined to take part in research activities [44]. Although statistical differences were observed, these should be interpreted with caution due to the sample imbalance.

A limitation of the research is that the newly created part of the questionnaire was reviewed for content relevance by only three independent experts. While this provides an initial indication of content validity, the small number of reviewers means that the results should be carefully interpreted [31].

## 5. Conclusions

This study highlights how nurses perceive the importance of continuing education. The results confirm that nurses with higher education levels and more experience tend to have greater motivations and a positive attitude toward career advancement. Previous engagement in higher education and an enthusiasm for further studies significantly influence nurses’ intentions to continue their education.

To motivate nurses to pursue ongoing education, it is essential to establish better working conditions, improve economic status, and create a supportive work environment. Opportunities for promotion and greater recognition within the profession can encourage nurses to invest in their professional development. Therefore, supporting access to educational opportunities, particularly for those who have already expressed interest, could enhance competencies and facilitate career development and advancement. Future research ought to investigate particular interventions or approaches that can help nurses in developing and sustaining motivating attitudes toward continuing education.

## Figures and Tables

**Table 1 nursrep-15-00190-t001:** Sociodemographic characteristics, work motivations, and attitudes toward further education of the nurses (N = 526).

	Not Intend to Enroll in Studies (n = 250)	Intend to Enroll in Studies (n = 276)	Test Statistic	*p*
Gender, (N, %)				
Female	234 (93.6)	249 (90.2)	2.00 ‡	0.157 *
Male	16 (6.4)	27 (9.8)		
Age (years), Mdn (IQR)	38.0 (13)	35.0 (14)	−7.77 §	<0.001
Environment, (N, %)				
Urban	200 (80.0)	223 (80.8)	0.05 ‡	0.818 *
Rural	50 (20.0)	53 (19.2)		
Family environment, (N, %)				
Live alone	11 (4.4)	22 (8.0)	2.85 ‡	0.092 *
Live with family, partner, roommate	239 (95.6)	254 (92.0)		
Have children, (N, %)				
Yes	185 (74.0)	150 (54.3)	21.91 ‡	<0.001 *
No	65 (26.0)	126 (45.7)		
Education level, (N, %)				
High school	98 (39.2)	113 (40.9)	9.75 ‡	0.021 *
Bachelor	88 (35.2)	121 (43.8)		
Master’s	58 (23.2)	37 (13.4)		
PhD	6 (2.4)	5 (1.8)		
Time from finish of high school (years), Mdn (IQR)	16.5 (15.0)	14.0 (17.0)	−7.42 §	<0.001 †
Length of service (years), Mdn (IQR)	20.0 (14.0)	16.0 (16.0)	−7.33 §	<0.001 †
Workplace, (N, %)				
Health center	39 (15.6)	48 (17.4)		0.004 *
Hospital/clinic	149 (59.6)	191 (69.2)	11.18 ‡	
Other institution	62 (24.8)	37 (13.4)		
Tried to enroll in studies before (N, %)				
Yes	170 (68.0)	245 (88.8)	33.98 ‡	<0.001 *
No	80 (32.0)	31 (11.2)		
Attitudes toward doctoral nursing studies (N, %)				
Positive	75 (30.0)	132 (47.8)	17.47 ‡	<0.001 *
Negative	175 (70.0)	144 (52.2)		

Note: * *p*-value Chi-square test; † *p*-value Mann-Whitney U Test; Mdn = median; IQR = Interquartile Range; Test-statistic (‡ = χ² value from Chi-square test, § = Z value from Mann–Whitney U test).

**Table 2 nursrep-15-00190-t002:** Differences in intrinsic and extrinsic motivational work orientations and attitudes toward further education between study groups (N = 526).

		Not Intend to Enroll in Studies (n = 250)	Intend to Enroll in Studies (n = 276)	Z	*p* *
Intrinsic motivation	Mdn (IQR)	60.0 (10.0)	62.0 (9.0)	−3.57	<0.001
	Mean Rank	238.7	286.0		
Challenge	Mdn (IQR)	26.0 (6.0)	27.0 (6.0)	−3.87	<0.001
	Mean Rank	236.6	287.8		
Enjoyment	Mdn (IQR)	34.0 (5.0)	35.0 (5.0)	−2.24	0.025
	Mean Rank	248.0	277.6		
Extrinsic motivation	Mdn (IQR)	51.0 (10.0)	51.0 (9.8)	−0.87	0.387
	Mean Rank	257.5	269.0		
Outward	Mdn (IQR)	35.0 (7.3)	34.0 (7.0)	−0.34	0.736
	Mean Rank	265.8	261.4		
Compensation	Mdn (IQR)	16.0 (4.0)	16.0 (3.8)	−2.41	0.016
	Mean Rank	246.8	278.6		
Overall work motivation (WPI)	Mdn (IQR)	110.5 (16.0)	113.0 (15.0)	−2.70	0.007
	Mean Rank	244.8	280.5		
Attitudes toward further education	Mdn (IQR)	33.0 (10.0)	40.5 (9.0)	−11.19	<0.001
	Mean Rank	185.7	334.0		

Note: * Mann–Whitney Test; Mdn = median; IQR = Interquartile Range; WPI = Work Preference Inventory Scale.

**Table 3 nursrep-15-00190-t003:** Associations between sociodemographic characteristics, motivational work orientations, and attitudes toward further education using linear regression models (N = 526).

	Extrinsic Motivation				Intrinsic Motivation				Overall Work Motivation		Positive Attitude Toward Further Education	
	Outward		Compensation		Challenge		Enjoyment					
	β	*p*	β	*p*	β	*p*	β	*p*	β	*p*	β	*p*
Age (years)	0.27	0.278	0.35	0.109	−0.68	0.004	−0.41	0.083	−0.33	0.159	−0.09	0.695
Gender (female was referent group)												
Male	−0.07	0.110	0.06	0.100	−0.04	0.369	−0.14	0.002	−0.12	0.005	−0.04	0.345
Environment (rural was referent group)												
Urban	0.02	0.636	0.86	0.942	0.01	0.818	0.04	0.384	0.03	0.485	0.02	0.665
Family environment (live with family, partner, or roommate is referent group)												
Live alone	−0.06	0.205	0.38	0.111	0.08	0.082	0.03	0.453	0.02	0.608	−0.06	0.160
Children (no was referent group)												
Have children	−0.05	0.373	0.96	0.706	0.03	0.517	−0.03	0.609	−0.02	0.695	−0.14	0.005
Education (university was referent group)												
High school	−0.12	0.006	0.01	0.095	−0.11	0.017	−0.08	0.087	−0.15	<0.001	−0.09	0.047
Time from finish of high school (years)	−0.04	0.814	0.53	0.502	0.24	0.169	−0.02	0.916	0.09	0.621	−0.10	0.586
Workplace (health centers were referent group)												
Working at hospital/clinic	−0.08	0.173	0.23	0.412	−0.15	0.005	−0.15	0.006	−0.17	0.002	−0.10	0.081
Working at other institution	0.03	0.602	0.68	0.808	−0.08	0.174	−0.02	0.729	−0.01	0.791	−0.10	0.081
Length of service (years)	−0.30	0.088	0.58	0.131	0.46	0.007	0.28	0.104	0.15	0.369	0.10	0.552
Attitudes toward doctoral nursing studies (negative attitude was referent group)												
Positive attitude toward doctoral studies	0.04	0.341	0.02	0.030	0.19	<0.001	0.13	0.003	0.16	<0.001	0.21	<0.001
Positive attitude toward further education	0.07	0.099	0.00	<0.001	0.13	0.004	0.08	0.087	0.16	<0.001	-	-
Adjusted R^2^	0.25		0.32		0.34		0.32		0.39		0.32	

Note: β = beta coefficient; *p* = *p*-value; adjusted R^2^ = adjusted explained variance.

**Table 4 nursrep-15-00190-t004:** Logistic regression model of characteristics associated with nursing study enrollment, N = 526.

	OR	95% CI	*p*
Gender (female was referent group)			
Male	1.32	(0.58–3.00)	0.509
Age (years)	0.93	(0.81–1.07)	0.303
Environment (rural was referent group)			
Urban	1.14	(0.66–1.99)	0.637
Family environment (Live with family, partner, or roommate is referent group)			
Live alone	1.56	(0.58–4.20)	0.379
Children (no was referent group)			
Have children	1.36	(0.78–2.38)	0.280
Education (university was referent group)			
High school	2.30	(1.29–4.10)	0.005
Time from finish of high school (years)	0.98	(0.89–1.08)	0.732
Workplace (health center is referent group)			
Working at hospital/clinic	1.37	(0.74–2.57)	0.319
Working at other institution	0.51	(0.24–1.08)	0.079
Length of service (years)	1.00	(0.91–1.10)	0.960
Tried to enroll in studies before (no was referent group)			
Enrolled in studies before	3.89	(1.97–7.67)	<0.001
Attitudes toward doctoral nursing studies (negative attitude was referent group)			
Positive attitude toward doctoral studies	1.78	(1.09–2.91)	0.022
Motivation			
Extrinsic motivation factors—outward	0.97	(0.92–1.02)	0.199
Extrinsic motivation factors—compensation	1.03	(0.94–1.12)	0.541
Intrinsic motivation factors—challenge	1.07	(1.01–1.14)	0.036
Intrinsic motivation factors—enjoyment	0.98	(0.92–1.05)	0.593
Attitudes toward further education	1.17	(1.13–1.22)	<0.001

Note: OR—odds ratio; 95% CI—95%confidence interval; *p*-value.

## Data Availability

The data presented in this study are available upon reasonable request from the corresponding author.

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
