# Peer review of "Motivations and Attitudes Toward Further Education: A Cross-Sectional, Descriptive Predictive Study"

_nursrep, 2025, doi:10.3390/nursrep15060190_

Round 1
Reviewer 1 Report
Comments and Suggestions for Authors
Global Appreciation
The article addresses the theme of continuing education in nursing as highly relevant, particularly in light of the increasing demands of the healthcare system and the ageing population. Regarding the methodology, the study employs a clearly described cross-sectional design with a substantial sample size (n=526), which enhances the statistical power of the analyses, thereby ensuring robustness.
In terms of measurement instruments, the use of validated scales such as the Work Preference Inventory and the Attitudes and Educational Intentions Scale is appropriate and supported by the literature. Ethical principles are duly observed, with adherence to the Declaration of Helsinki, and the study includes ethical approval, informed consent, and guarantees of anonymity.
The results are presented with clarity, with well-organised tables and statistically sound analyses, employing appropriate tests such as the Mann-Whitney test and logistic regression. The discussion is comprehensive, well-supported by literature, and successfully connects the findings to previous studies, offering both practical implications and policy suggestions.
Improvement Proposals
Some redundancy is noted in the title, particularly in the semantic construction "Motivations and Attitude to Towards Further Education." A reformulation is recommended to improve clarity and precision.
Several linguistic inconsistencies are identified, and the manuscript requires thorough English language editing. There are grammatical errors and unclear phrases, such as “No Intends Enrolling,” which should be corrected to “Do Not Intend to Enroll.” A professional language review is strongly advised.
It is recommended that the limitations currently discussed in the final section be reorganised into a separate section titled “Limitations”, as per MDPI submission guidelines.
Regarding the references, some appear duplicated or incorrectly formatted (e.g., references 4 and 5). Please ensure that all URLs and DOIs are current and fully functional.
In terms of MDPI formatting, several fields such as “Received,” “Accepted,” “Published,” and “Citation” are incomplete. While these can be finalised during the production stage, placeholders should be clearly indicated upon submission.
The discussion around male motivation should be approached with greater caution, considering the small proportion of male participants in the sample. Gender-based conclusions should be presented conservatively to avoid statistical overinterpretation.
Comments on the Quality of English Language
I suggest that the entire article be reviewed by a professional.
Author Response
Dear Sir or Madam, we would like to express our gratitude for your thoughtful and constructive review of our manuscript. Your detailed and well-structured comments were extremely valuable to us and helped us significantly improve the paper's clarity, precision, and overall quality. We especially appreciate your recognition of the topic's relevance and the strengths of our methodological approach. Please find our detailed responses below.
Comment 1:
Some redundancy is noted in the title, particularly in the semantic construction "Motivations and Attitude to Towards Further Education." A reformulation is recommended to improve clarity and precision.
Response to comment 1:
Thank you for this helpful observation. In response to your suggestion, we have revised the title to improve clarity and eliminate redundancy. The new title is:
“Motivations and Attitudes Towards Further Education Among Nurses: A Cross-Sectional Study.” We believe that this revised version more accurately reflects the content of the study while maintaining a clear and concise formulation.
Comment 2:
Several linguistic inconsistencies are identified, and the manuscript requires thorough English language editing. There are grammatical errors and unclear phrases, such as “No Intends Enrolling,” which should be corrected to “Do Not Intend to Enroll.” A professional language review is strongly advised.
Response to comment 2:
Thank you for pointing this out. Following your recommendation, we carefully revised the manuscript to correct all identified linguistic inconsistencies and grammatical errors. The phrase “No Intends Enrolling” has been corrected as suggested, along with other unclear expressions. The entire manuscript has undergone a thorough language revision by english teacher, and all changes have been clearly marked using track changes and highlighted in yellow for your convenience.
Comment 3:
It is recommended that the limitations currently discussed in the final section be reorganised into a separate section titled “Limitations”, as per MDPI submission guidelines.
Response to comment 3:
Thank you for this helpful suggestion. In accordance with the MDPI submission guidelines and your recommendation, we have created a separate section titled “Limitations” and moved the relevant content there. We believe this reorganisation improves the structure and clarity of the manuscript.
Comment 4:
Regarding the references, some appear duplicated or incorrectly formatted (e.g., references 4 and 5). Please ensure that all URLs and DOIs are current and fully functional.
Response to comment 4:
Thank you for your observation. We have carefully reviewed and corrected all references in the manuscript. For reference 4, since the cited Croatian journal does not provide a DOI, we included the URL and noted that the article is “available at” the respective website. For reference 5, which refers to data retrieved from a website, we added the access date in accordance with proper citation standards for online sources. All other URLs and DOIs have been verified and updated to ensure they are current and fully functional.
Comment 5 :
In terms of MDPI formatting, several fields such as “Received,” “Accepted,” “Published,” and “Citation” are incomplete. While these can be finalised during the production stage, placeholders should be clearly indicated upon submission.
Response to comment 5:
Thank you for this reminder. We have now inserted appropriate placeholders for the fields “Received,” “Accepted,” “Published,” and “Citation” in accordance with MDPI formatting guidelines. These have been clearly marked in the manuscript to indicate that they will be completed during the production stage.
Comment 6:
The discussion around male motivation should be approached with greater caution, considering the small proportion of male participants in the sample. Gender-based conclusions should be presented conservatively to avoid statistical overinterpretation.
Response to comment 6:
Thank you for this important observation. We agree that the small number of male participants limits the generalisability of gender-related findings. While gender-based statistical differences were reported in the results section, we did not interpret or emphasise these findings in the discussion. To address your concern, we have added a sentence in the discussion explicitly stating that the results related to male participants should be interpreted with caution due to their small representation in the sample (lines 336-338). Furthermore, we have revised the Limitations section to reflect this more clearly and included a reference to existing literature indicating that male individuals may be less inclined to participate in research studies (lines 366-367 and 368-369). We hope that this addition sufficiently addresses your valuable comment.
We hope that we have addressed all the comments appropriately, and we thank you in advance.
Kind regards,
The Authors
Reviewer 2 Report
Comments and Suggestions for Authors
Dear authors,
The topic is of relevance.
Lines 46 useful information about nurses education and training is added
Line 57 relevance is well explained
Line 77 obstacles, perhaps a wider range of authors on the obstacles can be added
Line 81 two types of motivation: intrinsic and extrinsic, perhaps definition or examples can include awards, recognition; as lines 127-139 more factors are recognized and put into the survey
Keywords: there are no job-related key words; based on research and findings it feels like they may be added
Line 103 Based on "intention to enrol in studies" – what is a question in the survey?
Line 366 This study highlights the importance of continuing education for nurses. – perhaps the study highlights how nurses envision the importance of continuing education. One of recommendations within prolongation of your study can be certain measures to help nurses to shift into motivational attitude towards continuing education.
Author Response
Dear sir or madam, thank you very much for your thoughtful reading and positive comments regarding the relevance of the topic and the clarity of the information provided. We appreciate your feedback and have marked all changes in the manuscript using track changes and highlighted them in yellow to facilitate review and ensure easier navigation through the revised text.
Comment 1:
Line 77 obstacles, perhaps a wider range of authors on the obstacles can be added
Response to comment 1:
Thank you for your suggestion. In the manuscript, we referenced two relevant sources discussing the obstacles to continuing education among nurses. After reviewing the available literature, we found that most studies consistently identify the same or very similar types of barriers/obscales. For that reason, we chose to include only the most relevant and representative sources to avoid unnecessary repetition. We hope that this focused selection will be acceptable and sufficient for the purpose of this section.
Comment 2:
Line 81 two types of motivation: intrinsic and extrinsic, perhaps definition or examples can include awards, recognition; as lines 127-139 more factors are recognized and put into the survey
Response to comment 2:
Thank you for this helpful comment. We have revised the sentence to include brief explanations and examples of intrinsic and extrinsic motivation, as suggested. The revised version (lines 84-85) of the sentence is as follows: “It can be intrinsic motivation, based on greater interest and satisfaction (e.g., personal satisfaction), or extrinsic motivation, based on the expected outcome (e.g., awards, recognition).”
Comment 3:
Keywords: there are no job-related key words; based on research and findings it feels like they may be added
Response to comment 3:
Thank you for your observation. In response to your suggestion, we have revised the list of keywords by replacing “academic success” with “professional development,” which more accurately reflects the job-related focus of the study.
Comment 4:
Line 103 Based on "intention to enrol in studies" – what is a question in the survey?
Response to comment 4:
Thank you for your comment. We have clarified this point in the manuscript. The division of participants into two study groups was based on the following survey question:
“Do you intend to enrol in further nursing education?” Participants answered with either Yes or No, which served as the basis for group categorisation. This clarification has been added in lines 111–114 of the revised manuscript.
Comment 5:
Line 366 This study highlights the importance of continuing education for nurses. – perhaps the study highlights how nurses envision the importance of continuing education. One of recommendations within prolongation of your study can be certain measures to help nurses to shift into motivational attitude towards continuing education.
Response to comment 5:
Thank you for this insightful suggestion. We have revised the sentence to clarify that the study highlights how nurses perceive the importance of continuing education. Additionally, in response to your comment, we have added a concluding sentence recommending that future research explore specific strategies to support the development of a motivational attitude towards lifelong learning among nurses. We believe this addition strengthens the relevance and applicability of our findings.
We hope that we have addressed all the comments appropriately, and we thank you in advance.
Kind regards,
The Authors
Reviewer 3 Report
Comments and Suggestions for Authors
Dear Authors,
Thank you for allowing me to review your manuscript. Your work addresses a highly relevant topic, and I find that part of the results are particularly interesting within the international context. The manuscript is sufficiently clear and accurate; however, I would suggest a few minor revisions to improve its clarity.
-
LINE 91 – You mention the use of an online questionnaire; please specify the online platform used to create the online version of the questionnaire.
-
LINE 98 – You wrote: "A limit was set for completing the survey from the same IP address to avoid duplicate responses." I believe you meant a "time limit". Please clarify this more precisely, if possible.
-
LINES 107–108 – You refer to a "newly constructed questionnaire," but from lines 122–123, it appears that part of the questionnaire (the WIPI) has already been used in previous research. As currently written, this is somewhat confusing. Please clarify already in lines 107–108 that the questionnaire consists of three parts: one socio-demographic section, one previously validated tool (the WIPI – please specify whether it was validated), and one newly developed section.
-
LINES 122–124 / 144–145 – Reliability results of the instruments are presented, but this is still part of the Methods section. Please move these results to the RESULTS section.
-
LINES 126–145 – The draft of the new questionnaire is said to be based on a literature review. However, it is unclear how many authors participated in the drafting process, whether a Content Validity Index (CVI) was calculated to assess content validity, and how this impacted the study. If a CVI was not conducted, this represents a significant LIMITATION of the study and should be discussed in the Limitations section. The absence of a CVI likely contributes to the low Cronbach’s Alpha values. Considering the factors that influence this indicator, a value of 0.6 is rather low.
-
LIMITATIONS – This section needs to be significantly expanded. Beyond the missing CVI and its implications, the limitations of snowball sampling—particularly how it restricts the generalizability of findings—should be discussed. Furthermore, the lack of validation of the newly developed instrument must be carefully considered when interpreting the results, especially given the low Cronbach’s Alpha.
Good luck for your work!
Author Response
Dear sir or madam, we sincerely thank you for your time, effort, and valuable comments regarding our manuscript. We truly appreciate your detailed and thoughtful review, which helped us identify important areas for clarification and improvement. Your suggestions have significantly contributed to enhancing the quality, accuracy, and transparency of our work. We have carefully considered each of your comments and revised the manuscript accordingly. Below, we provide our point-by-point responses, with all changes clearly marked in the revised manuscript using track changes and highlighted in yellow to facilitate review.
Comment 1:
LINE 91 – You mention the use of an online questionnaire; please specify the online platform used to create the online version of the questionnaire.
Response to comment 1:
Thank you for your comment. To clarify this point, we have added the following sentence to the manuscript: “Google Forms platform was used for making the online survey.” This sentence was inserted for clarity in lines 98–99 of the revised manuscript.
Comment 2:
LINE 98 – You wrote: "A limit was set for completing the survey from the same IP address to avoid duplicate responses." I believe you meant a "time limit". Please clarify this more precisely, if possible.
Response to comment 2:
Thank you for your observation. We confirm that the intention was not to set a time limit, but rather to restrict multiple submissions from the same IP address in order to prevent duplicate responses. We have rephrased the sentence in the manuscript for clarity. The revised (line 105) sentence is as follows: “To prevent duplicate responses, only one submission per IP address was allowed.”
Comment 3:
LINES 107–108 – You refer to a "newly constructed questionnaire," but from lines 122–123, it appears that part of the questionnaire (the WIPI) has already been used in previous research. As currently written, this is somewhat confusing. Please clarify already in lines 107–108 that the questionnaire consists of three parts: one socio-demographic section, one previously validated tool (the WIPI – please specify whether it was validated), and one newly developed section.
Response to comment 3:
Thank you for this helpful observation. We agree that the original formulation may have been misleading, as it implied that the entire questionnaire was newly constructed. To improve clarity, we have revised the introductory sentence in lines 120–121 to describe the instrument as a structured questionnaire consisting of three parts, instead of referring to it as a newly constructed questionnaire. Additionally, we have clarified in the description of the third part of the questionnaire that this section was newly developed (lines 140-147). These changes more accurately reflect the composition of the instrument and eliminate ambiguity regarding its structure.
Comment 4:
LINES 122–124 / 144–145 – Reliability results of the instruments are presented, but this is still part of the Methods section. Please move these results to the RESULTS section.
Response to comment 4:
Thank you for your observation. We acknowledge that reliability coefficients are typically reported in the Results section. However, as the primary aim of this study was not to validate the instruments but to assess motivation and attitudes, we included Cronbach’s alpha values within the Methods section solely to inform readers about the internal consistency of the scales used in this specific context. We consider this information to be a methodological detail related to the description of the instruments, rather than a result per se. For this reason, we respectfully retained these values in the Methods section. We hope this approach will be acceptable.
Comment 5:
LINES 126–145 – The draft of the new questionnaire is said to be based on a literature review. However, it is unclear how many authors participated in the drafting process, whether a Content Validity Index (CVI) was calculated to assess content validity, and how this impacted the study. If a CVI was not conducted, this represents a significant LIMITATION of the study and should be discussed in the Limitations section. The absence of a CVI likely contributes to the low Cronbach’s Alpha values. Considering the factors that influence this indicator, a value of 0.6 is rather low.
Response to comment 5:
Thank you for this thoughtful and important observation. We have now clarified in the Methods section (lines 140-147) that this part of the instrument was developed by the first and last authors, based on a review of the relevant literature and professional experience of the last author. The content was then independently reviewed by three experts in nursing with extensive clinical, academic, and educational backgrounds. Their feedback was used to refine the clarity and relevance of the questionnaire items.
Comment 6:
LIMITATIONS – This section needs to be significantly expanded. Beyond the missing CVI and its implications, the limitations of snowball sampling—particularly how it restricts the generalizability of findings—should be discussed. Furthermore, the lack of validation of the newly developed instrument must be carefully considered when interpreting the results, especially given the low Cronbach’s Alpha.
Response to comment 6:
Although a formal CVI calculation with a larger expert panel was not performed, this review represents a preliminary assessment of content validity. We have added this as a limitation in the revised manuscript, as recommended. This clarification also reflects the exploratory nature of the newly developed section. The relevant literature has been cited accordingly (https://doi.org/10.1002/nur.20147).
We hope that we have addressed all the comments appropriately, and we thank you in advance.
Kind regards,
The Authors